# Factors Affecting Nitrous Oxide Emissions from Activated Sludge Wastewater Treatment Plants—A Review

Liana Kemmou and Elisavet Amanatidou *

Laboratory of Environmental Chemistry & Water and Wastewater Treatment, Department of Chemical Engineering, University of Western Macedonia, 50100 Kozani, Greece
* Correspondence: eamanatidou@uowm.gr; Tel.: +30-2461068015

**Abstract:** Nitrous oxide ($N_2O$) is a greenhouse gas contributing to ozone layer depletion and climate change. Wastewater treatment plants (WWTPs) contribute significantly to the global anthropogenic $N_2O$ emissions. The main factors affecting $N_2O$ emissions are the dissolved oxygen concentration (DO), the nitrite accumulation, the rapidly changing process conditions, the substrate composition and COD/N ratio, the pH, and the temperature. Low DO in the nitrification process results in higher $N_2O$ emissions, whereas high aeration rate in the nitration/anammox process results in higher $N_2O$ production. High DO in the denitrification inhibits the $N_2O$ reductase synthesis/activity, leading to $N_2O$ accumulation. High nitrite accumulation in both the nitrification and denitrification processes leads to high $N_2O$ emissions. Transient DO changes and rapid shifts in pH result in high $N_2O$ production. Ammonia shock loads leads to incomplete nitrification, resulting in $NO_2^-$ accumulation and $N_2O$ formation. Limiting the biodegradable substrate hinders complete denitrification, leading to high $N_2O$ production. A COD/N ratio above 4 results in 20–30% of the nitrogen load being $N_2O$ emissions. Maximum $N_2O$ production at low pH (pH = 6) was observed during nitrification/denitrification and at high pH (pH = 8) during partial nitrification. High temperature enhances the denitrification kinetics but produces more $N_2O$ emissions.

**Keywords:** $N_2O$ production pathways; hydroxylamine oxidation; nitrifier denitrification; heterotrophic denitrification; influence of DO; COD/N; pH and temperature; $N_2O$ sampling; $N_2O$ measurements

## 1. Introduction

Wastewater treatment plants (WWTPs) generate significant amounts of greenhouse gases, including carbon dioxide, methane and nitrous oxide [1]. According to the IPCC 2023, the global anthropogenic emissions of greenhouse gases (GHGs) were $59 \pm 6.6$ GtCO$_2$-eq, about 12% (6.5 GtCO$_2$-eq) higher than in 2010 and 54% (21 GtCO$_2$-eq) higher than in 1990 [2]. The contribution of each gas to the total GHG emissions varies: 79.4% for carbon dioxide, 11.5% for methane, and 6.2% for nitrous oxide, with the remainder consisting of fluorinated gases [3]. Nitrous oxide ($N_2O$) is an important greenhouse gas with a global warming potential (GWP) 273 times that of carbon dioxide ($CO_2$) [2], contributing to ozone layer depletion and climate change [4]. Global $N_2O$ emissions were $2.7 \pm 1.6$ GtCO$_2$-eq, according to the IPCC 2023 [2]. Therefore, even small amounts of $N_2O$ emissions can significantly contribute to total GHG emissions. Thus, it can be concluded that the minimization of $N_2O$ emissions and the identification of the factors controlling these emissions constitute a great challenge.

Human activities such as agriculture and fossil fuel combustion, along with microbial processes occurring in biological wastewater treatment are the main sources of anthropogenic $N_2O$ emissions [5–8]. Wastewater treatment as a sector contributes approximately 3% of the global anthropogenic $N_2O$ emissions [1,9–11]. This percentage has increased in recent years [12–14]. Yao et al., 2022 reported that wastewater treatment was the fourth-largest source of $N_2O$ emissions after agriculture, energy production, and other industrial

production activities, accounting for 5.6% of total $N_2O$ emissions. $N_2O$ emissions originating from sludge disposal and treatment are not included in those from the wastewater treatment sector. The total amounts of $N_2O$ produced from sludge incineration, reuse in cement, and composting are 645.0 kg $N_2O$/tonne, 294 kg $N_2O$/tonne and 0.37 kg $N_2O$/tonne, respectively [15]. $N_2O$ emission from wastewater management contributes 26% of the total GHGs originating from the water sector, which includes drinking water production, water transport, wastewater and sludge treatment and discharge [10,16].

Numerous studies have confirmed that in biological wastewater treatment, the nitrification and denitrification processes occurring under aerobic and anaerobic/anoxic conditions are generally responsible for $N_2O$ emissions [17–26]. The design criteria for conventional and extended aeration activated sludge (AS) processes that include nitrification and denitrification are presented in Table 1. Nitrification comprises two coupled processes (Figure 1): (1) ammonium ($NH_4^+$) is oxidized to hydroxylamine ($NH_2OH$) and nitrite ($NO_2$) by autotrophic ammonia-oxidizing bacteria (AOB) or by autotrophic ammonia-oxidizing archaea (AOA); (2) $NO_2^-$ is oxidized to nitrate ($NO_3^-$) by nitrite-oxidizing bacteria (NOB) through a reaction known as nitratation [27]. For many decades, it has been known that autotrophic aerobic bacteria are responsible for ammonia and nitrite oxidation. However, under aerobic conditions, a heterotrophic nitrification process can occur, in which inorganic and organic reduced forms of nitrogen are oxidized to nitrate by a wide range of fungi and heterotrophic bacteria [28]. The partial nitrification of ammonia to nitrite is known as nitritation, where the ammonia oxidation is controlled, leading to nitrite instead of nitrate by inhibiting or eliminating nitrite oxidizing bacteria (NOB) under low dissolved oxygen concentrations and high temperatures. During nitritation, ammonia-oxidizing bacteria (AOB) become the dominant nitrifying bacteria, and accumulation of nitrites is achieved [29].

**Table 1.** Design criteria for conventional and extended aeration activated sludge (AS) processes.

| Parameters | Typical Values | |
| --- | --- | --- |
| | Conventional AS Process | Extended Aeration AS Process |
| MLSS (mg/L) | 1500–4000 | 2000–6000 |
| F/M (kgBOD/day/kg MLSS) | 0.2–0.5 | 0.05–0.15 |
| HRT (hours) | 4–8 | 18–36 |
| SRT (days) | 5–10 | 20–40 |
| $C_v$ (kgBOD/day/m$^3$) | 0.4–0.8 | 0.15–0.25 |
| Wasted sludge (kg/kg BOD removed) | 0.4–0.6 | 0.15–0.3 |
| Sludge recycling (%) | 50–100 | 75–150 |

$$NH_4^+ \xrightarrow{AOB} NH_2OH \xrightarrow{AOB} NO_2^- \xrightarrow{NOB} NO_3^-$$

**Figure 1.** Nitrification process with ammonia-oxidizing bacteria (AOB) and nitrite-oxidizing bacteria (NOB).

Denitrification is the process by which $NO_3^-$ is reduced to $N_2$ by heterotrophic anaerobic bacteria using carbon sources as electron donors (Figure 2). The intermediate products of nitrate reduction are $NO_2^-$, NO, and $N_2O$ [30]. Moreover, heterotrophic denitritation can occur under anaerobic conditions, where nitrite is directly reduced to $N_2$ gas by heterotrophic anaerobic bacteria [21].

$$NO_3^- \xrightarrow{HB} NO_2^- \xrightarrow{HB} NO \xrightarrow{HB} N_2O \xrightarrow{HB} N_2$$

**Figure 2.** Denitrification process by heterotrophic bacteria (HB).

A partial nitritation/anammox process can occur in wastewater treatment processes under low oxygen aerobic conditions, where half of the $NH_4^+$ is oxidized to $NO_2^-$ by AOB

and the remaining $NH_4^+$, along with the produced $NO_2^-$, is directly converted to nitrogen gas by anammox bacteria [31]. The anammox bacteria were identified by Strous et al., 1999 as an autotrophic member of the order *Planctomycetales* [32]. Some anammox genera that have been discovered include fresh-water and marine species [33].

Many researchers have investigated the mechanisms of $N_2O$ production in WWTPs [5,25,34–41]. It is known that $N_2O$ formation pathways include hydroxylamine oxidation, nitrifier denitrification and heterotrophic denitrification [42–48]. The main mechanism favoring $N_2O$ production in WWTPs depends on the process configuration and operational parameters [49,50]. Goreau et al., 1980 concluded that the denitrification pathway of AOB was responsible for $N_2O$ emissions. In order to reveal possible mechanisms of $N_2O$ production, many factors affecting $N_2O$ emissions have been reported: low dissolved oxygen concentration in aerobic conditions or high dissolved oxygen concentration in anoxic conditions, accumulation of nitrite, rapidly changing process conditions, pH, temperature or a low ratio of COD to nitrogen compounds during heterotrophic denitrification [34,51–57].

The aim of this work is to provide a review, using a critical perspective, (a) the possible pathways of $N_2O$ production, in order to better understand the formation of $N_2O$ in different WWTPs, (b) the factors affecting $N_2O$ emissions, and (c) the techniques employed for the measurement of $N_2O$ emissions, and the sampling strategies used.

## 2. $N_2O$ Production Pathways

Many studies have been conducted since 1960 investigating the production mechanisms of $N_2O$ [42–45,58]. It is agreed that $N_2O$ can be produced biologically via three pathways: hydroxylamine oxidation, nitrifier denitrification and heterotrophic denitrification [5,6,42–47,58–61]. $N_2O$ can also be formed via abiotic pathways, but this contribution to overall $N_2O$ emissions is considered negligible [62]. The main difference among researchers lies in the possibility of the production of $N_2O$ through pathways related to hydroxylamine oxidation.

### 2.1. Hydroxylamine Oxidation

The oxidation of ammonium to nitrites under aerobic conditions involves a two-step process. In the first step, ammonium is converted to hydroxylamine, and in the second step, hydroxylamine is converted to nitrites through intermediate products. The first step is catalyzed by a membrane-bound ammonia mono-oxygenase (AMO), and the subsequent step is catalyzed by hydroxylamine oxidoreductase (HAO). HAO is an abundant periplasmic cytochrome *c* of nitrifying chemoautotrophic bacteria *Nitrosomonas europaea* [63].

Many studies have confirmed that $O_2$ and a pair of electrons are required for the 1st step [6,63,64] while two pairs of electrons are required for the second step. One pair is employed for the oxidation of ammonium to nitrites, and the remaining pair is employed for respiratory electron transfer, terminating in a cytochrome oxidase [63]. However, the complete catalytic cycle of hydroxylamine oxidation to nitrites remains unclear [65]. Different conclusions have been drawn by researchers regarding the intermediate products of the oxidation of hydroxylamine to nitrites.

Igarashi et al., 1997 [45] studied the HAO structure and proposed that the oxidation of hydroxylamine can be divided into two reactions facilitating the simultaneous acceptance and transmission of two electrons. The two reactions occur simultaneously. Hydroxylamine is converted into a nitrosyl HNO, and HNO is converted into nitrites. Immediate and continuous oxidation with two electrons is necessary to maintain the electron flow; otherwise, HAO produces $N_2O$ or NO from HNO. The intermediates and $N_2O$ formed during the oxidation of ammonium to nitrites reported by Igarashi et al., 1997 [45] are presented in Figure 3:

$$NH_3 \xrightarrow{AMO} NH_2OH \xrightarrow{HAO} HNO \longrightarrow NO_2^-$$
$$\downarrow$$
$$NO, N_2O$$

**Figure 3.** Formation of intermediates and $N_2O$ during the oxidation of $NH_3$ to $NO_2^-$, adapted from Igarashi et al., 1997 [45].

Chen et al., 2018 [37] agreed with the results of Igarashi et al., 1997 [45] regarding the formation of intermediates and $N_2O$ generation pathways during the ammonium oxidation to nitrites. They reported that, under ammonium-oxidizing conditions, $O_2$ or nitrites can be used as the electron acceptor for the oxidation of hydroxylamine to nitrous oxide. These reactions can take place only if hydroxylamine or nitrites have been accumulated (Figure 4). Guo et al., 2018 [39], in agreement with Igarashi et al., 1997 [45], concluded that when conditions are not favorable, hydroxylamine oxidoreductase catalyzes the conversion of hydroxylamine to nitrosyl (NOH), instead to nitrite. The subsequent polymerization and hydrolysis of NOH can yield $N_2O$ through the reaction of Equation (1). Caranto et al., 2016 reported that direct production of $N_2O$ from hydroxylamine by cytochrome *P460* (a *c*-type heme of hydroxylamine oxidoreductase) under anaerobic conditions can occur [66].

$$NH_2OH \longrightarrow 2NOH \longrightarrow N_2O_2H_2 \longrightarrow N_2O + H_2O \tag{1}$$

**Figure 4.** Formation of intermediates and $N_2O$ during the oxidation of $NH_3$ to $NO_2^-$, adapted from Chen et al., 2018 [37].

Conversely, Anderson 1964 [42] suggested that the metabolism of $NH_2OH$ was a two-stage process. In the first stage, $NH_2OH$ is dehydrogenated to form NOH (or HNO), which can be further dehydrogenated to NO or to $N_2O$, as described in Equations (2)–(4). In the second stage, NO is converted to $NO_2^-$ by an enzyme system requiring oxygen (Equation (5)).

$$NH_2OH \longrightarrow NOH + 2e^- + 2H^+ \tag{2}$$

$$NOH \longrightarrow e^- + H^+ + NO \tag{3}$$

$$NOH \longrightarrow 0.5N_2O + 0.5H_2O \tag{4}$$

$$NO + H_2O \longrightarrow NO_2^-, H^+ + e^- + H^+ \tag{5}$$

The intermediates and $N_2O$ formed during the oxidation of $NH_3$ to $NO_2^-$, according to Anderson 1964 [42], are presented in Figure 5. Anderson 1964 [42] concluded that NO was an obligatory intermediate in the oxidation of $NH_2OH$ to $NO_2^-$. Moreover, Poughon et al., 2001 [46] obtained NO and $N_2O$ through several enzymatic and nonenzymatic reactions and confirmed their production directly from the unstable intermediate HNO. They also showed that the production of $N_2O$ resulted from the nonenzymatic dehydrogenation of unstable HNO in accordance with Equation (4).

$$NH_3 \xrightarrow{AMO} NH_2OH \xrightarrow{HAO} HNO \rightarrow NO \rightarrow NO_2^-$$
$$\downarrow$$
$$N_2O$$

**Figure 5.** Formation of intermediates and $N_2O$ during the oxidation of $NH_3$ to $NO_2^-$, adapted from Anderson 1964 [42].

Hooper and Terry 1979 [65], utilizing a sample of purified HAO with low nitrite reductase activity, showed that NO was a product of the hydroxylamine oxidation and was not produced by the reduction of $NO_2^-$. The possible inhibition of the HNO to NO oxidation results in the accumulation of $N_2O$. The presence of Mn(II) results in decreased formation of NO and increased formation of $N_2O$. The separate experimental steps of $NH_2OH$ oxidation are presented in Equations (6)–(8):

$$NH_2OH \longrightarrow HNO \tag{6}$$

$$HNO \longrightarrow NO \text{ (blocked by Mn)} \tag{7}$$

$$NO \longrightarrow HNO_2 \text{ (blocked anaerobically)} \tag{8}$$

Caranto and Lancaster 2017 [35] showed that nitric oxide was not merely a by-product, but rather an obligatory intermediate of the ammonium oxidation to nitrites. They presented a new model for AOB metabolism that implied that there were two obligatory intermediates (hydroxylamine and nitric oxide) during the ammonium oxidation by AOB, and that a mediator was required for the third enzymatic step (Figure 6). They provided evidence that hydroxylamine oxidoreductase oxidizes hydroxylamine with only three electrons to nitric oxide both aerobically and anaerobically. Nitrite found that hydroxylamine oxidoreductase activity assays is a nonenzymatic product resulting from the aerobic oxidation of nitric oxide by $O_2$.

$$NH_3 \xrightarrow{AMO} NH_2OH \xrightarrow{HAO} NO \longrightarrow NO_2^-$$
$$\downarrow$$
$$N_2O$$

**Figure 6.** Formation of intermediates and $N_2O$ during the oxidation of $NH_3$ to $NO_2^-$, adapted from Caranto and Lancaster 2017 [35].

As pointed out by Ritchie and Nicholas 1972 [43], NO cannot be considered an obligatory intermediate until the enzyme-catalyzed conversion of NO to $NO_2^-$ has been demonstrated. Yoshida and Alexander 1970 and Ritchie and Nicholas 1972 agreed that $N_2O$ was produced by washed cells of Nitrosomonas during the oxidation of ammonia and $NH_2OH$ [43,67]. They also indicated $N_2O_2H_2$ to be an intermediate of the oxidation of hydroxylamine (Figure 7).

$$NH_3 \xrightarrow{AMO} NH_2OH \xrightarrow{HAO} NOH \longrightarrow N_2O_2H_2 \longrightarrow NO \longrightarrow NO_2^-$$
$$_{+O_2}\swarrow \qquad \searrow_{+NO_2^-} \qquad \downarrow$$
$$N_2O \qquad N_2O \qquad N_2O$$

**Figure 7.** Formation of intermediates and $N_2O$ during the oxidation of $NH_3$ to $NO_2^-$, adapted from Ritchie and Nicholas 1972 [43].

The $N_2O$ production pathways studied by Wunderlin et al., 2012 [5] are in agreement with Ritchie and Nicholas 1972 [43]. Wunderlin et al., 2012 carried out batch tests and

reported that $N_2O$ production occurred during biological hydroxylamine oxidation, and intermediates such as HNO or $N_2O_2H_2$ are directly involved in its formation [43,46]. Wunderlin et al., 2012 [5] also investigated chemical $N_2O$ production. Tests were conducted with the addition of hydroxylamine to tap $H_2O$. Low amounts of $N_2O$ were formed under oxic conditions but were slightly increased when nitrites were added. Thus, hydroxylamine chemical decomposition and a chemical reaction between hydroxylamine and nitrites could be potential sources of $N_2O$ formation. The proposed mechanism is presented in Equations (9) and (10) [43,68]:

$$NH_2OH + 0.5O_2 \longrightarrow 0.5N_2O + 1.5H_2O \tag{9}$$

$$NH_2OH + NO_2^- + H^+ \longrightarrow N_2O + 2H_2O \tag{10}$$

Massara et al., 2018 [38] described the $N_2O$ emissions in municipal WWTPs by developing an ASM2d-$N_2O$ model. The formation of intermediates and $N_2O$ during the oxidation of $NH_3$ to $NO_2^-$ are summarized below (Figure 8):

**Figure 8.** Formation of intermediates and $N_2O$ during the oxidation of $NH_3$ to $NO_2^-$, adapted from Massara et al., 2018 [38].

Wan et al., 2019 [41] set up a one-stage partial nitrification process (nitritation-anammox reactor), which was in agreement with the $N_2O$ production pathways described by Massara et al., 2018 [38]. Wan et al., 2019 [41] also reported the production of $N_2O$ by NOB with pyruvate as an electron donor.

## 2.2. Heterotrophic Denitrification

$N_2O$ is an intermediate for the heterotrophic denitrification pathway. During denitrification, nitrates are reduced to nitrites catalyzed by nitrate reductase (NaR), then nitrites to nitric oxide catalyzed by nitrite reductase (NiR), nitric oxide to nitrous oxide catalyzed by nitric oxide reductase (NOR), and, lastly, nitrous oxide to diatomic molecule nitrogen catalyzed by nitrous oxide reductase ($N_2OR$) [28,28,59]. During complete denitrification, consumption of $N_2O$ is observed.

## 2.3. Nitrifier Denitrification

AOB contain a periplasmic copper containing NiR and NOR [69,70]. NiR could speed up $NH_2OH$ oxidation by channeling electrons from the cytochrome pool to $NO_2^-$ (to form NO), thus playing a facilitative role in $NH_3$ oxidation itself [69,70]. AOB also possess the inventory to alternatively convert NO into $N_2O$, using a haem-copper nitric oxide reductase, sNOR [70]. Thus, during nitrifier denitrification, nitrites are reduced to NO, NO is reduced to $N_2O$, and then $N_2O$ is reduced to diatomic molecular nitrogen by autotrophic ammonium-oxidizing bacteria. However, only genes encoding nitrite and nitric oxide reductases are found in the ammonium-oxidizing bacteria genome, and not nitrous oxide reductase. This means that nitrous oxide, rather than diatomic molecular nitrogen, is the end product of the nitrifier denitrification pathway [71]. Many researchers have suggested that $N_2O$ production can be attributed to a pathway involving nitrifier denitrification by ammonia-oxidizing bacteria, especially under anoxic conditions [49,51,71–74]. Kim et al., 2010 also demonstrated that nitrifier denitrification by ammonia-oxidizing bacteria is a major source for nitrous oxide production in an activated sludge under nitrifying conditions.

### 2.4. Abiotic Pathways

Hydroxylamine and nitrites are intermediates during the oxidation of ammonium to nitrates (nitrification), and they can be involved in the chemical reactions that yield $N_2O$, especially in the presence of trace metals [62]. Hydroxylamine can be oxidized by oxygen or react with nitrous acid ($HNO_2$) to generate $N_2O$ in abiotic conditions [75]. The contribution of this abiotic pathway to $N_2O$ production is not considered negligible in several nitrogen removal processes, e.g., the partial nitrification (nitritation) process [75]. Hydroxylamine can be oxidized by Fe (III) to form $N_2O$, but in this reaction, the formation of $N_2O$ can occur in acidic pH and in soils rather than in WWTPs [76]. Ferrous iron [Fe (II)] can reduce nitrites to nitric oxide, and in the second reaction step, nitric oxide to $N_2O$ [76]. The contribution of this abiotic pathway to $N_2O$ production is significant in wastewater treatment only if nitrites and Fe (II) are present concomitantly [77]. $N_2O$ can be abiotically formed by the degradation of hydroxylamine in the presence of air and low concentrations of copper [78]. Although in most previous studies the contribution of these chemical reactions was considered negligible or unimportant, they are likely to be enhanced in wastewater containing heavy metals [79].

## 3. Factors That Affect $N_2O$ Emissions

According to different $N_2O$ production pathways, the main factors affecting $N_2O$ emissions are the dissolved oxygen concentration (DO), the nitrite accumulation, the rapidly changing process conditions (e.g., high ammonia concentration and oxygen limitation), the substrate composition and COD/N ratio, the pH, and the temperature. The climatic zone, the location, the performance, and the influent characteristics of WWTPs also influence $N_2O$ emissions. These factors are dependent on the various microorganism species (AOB, NOB, AOA), whereas the microorganisms present in each process are dependent on the substrate and process conditions.

### 3.1. Dissolved Oxygen Concentrations (DO)

The dissolved oxygen concentration is considered a very important parameter controlling $N_2O$ emissions during nitrification (nitration) or nitritation. Low DO concentrations during nitrification result in high $N_2O$ emissions, which can be attributed to nitrifier denitrification (see Section 2.3) [49,51,52,55,80–82]. Li et al., 2015 investigated the synergistic effect of DO and pH on $N_2O$ emissions in a pilot-scale SBR process and reported that when DO was decreased from 3 to 0.5 mg/L, more $NO_2^-$ was accumulated, resulting in $N_2O$ production (nitrifier denitrification). Similar observations were made by Zheng et al., 1994 reporting that at DO < 1 mg/L $N_2O$ production increased, due to nitrifier denitrification. A weak negative correlation between DO concentration and dissolved $N_2O$ emissions (Spearman's correlation coefficient equal to $-0.7$) was confirmed by Vasilaki et al., 2020. In addition to nitrifier denitrification, the hydroxylamine oxidation pathway was found to contribute to $N_2O$ emissions in the study of Peng et al., 2014. They demonstrated that the specific $N_2O$ production rate increased from 0 to 1.9 mgN$_2$O-N/h/grVSS when DO concentration was increased from 0 to 3 mg/L [83]. With an increase in DO from 0.2 to 3 mg/L, the contribution of nitrifier denitrification by AOB decreased from 92% to 73%, accompanied by a corresponding increase in the contribution by the hydroxylamine oxidation pathway [83]. The transition from anoxic to aerobic conditions resulted in the accumulation of hydroxylamine and the formation of $N_2O$ through the hydroxylamine oxidation pathway [70]. High $N_2O$ production was observed under an increased aeration rate in a partial nitritation anammox reactor [31,41,84]. The stronger aeration, accompanied by an increased DO, stimulates stripping, leading to an increased proportion of the produced $N_2O$ leaving via the gas phase [41].

Dissolved oxygen affects $N_2O$ production during denitrification by inhibiting the synthesis and activity of nitrous oxide reductase, and its activity has been found to stop immediately when the denitrifying bacteria move from an anaerobic to an aerobic environ-

ment [28]. Nitrite reductase activity continues at a lower rate under the same transition, so that nitrous oxide emissions will occur [6].

### 3.2. Nitrite Accumulation

Nitrites are formed by AOB ammonium oxidation and by the reduction of heterotrophic bacteria nitrates. $NO_2^-$ plays a key role in nitrous oxide production. $NO_2^-$ accumulation increases nitrous oxide emissions during nitrification and denitrification.

During nitrification, increased nitrite concentrations can lead to increased nitrifier denitrification by AOB and increased $N_2O$ emissions [6,17,34,54,56,57,71,85]. High nitrite concentrations and low DO concentrations are known triggers for nitrite reductase and nitric oxide reductase expression in AOB, which favors $N_2O$ production through the nitrifier denitrification pathway [86].

During nitritation–denitritation applying SBR process, increased nitrous oxide emissions were observed at high $NO_2^-$ concentrations and DO < 1.5 mg/L [87]. This was related to a promoted expression of nitric oxide reductase gene or increased activity of $NO_2^-$ reductase with increasing substrate concentration [88]. Thus, at lower DO (<1.5 mg/L), AOB denitrification was possibly responsible for $N_2O$ production during nitritation.

In heterotrophic denitrification, high $NO_2^-$ concentrations inhibit complete denitrification, resulting in nitric oxide and nitrous oxide accumulation and increased nitrous oxide emissions [24,89–93]. Under elevated $NO_2^-$ concentrations, NiR, NOR and $N_2OR$ compete for electrons [94]. Limited generation of nitric oxide reductase under high concentrations of $NO_2^-$ during denitrification has also been observed, resulting in NO accumulation [89]. This can further affect nitrous oxide emissions, as nitric oxide inhibits the activity of the enzymes involved in the denitrification process.

### 3.3. Rapidly Changing Process Conditions

In many studies, elevated nitrous oxide emissions were reported when the process conditions were changed rapidly (e.g., high ammonia concentration and oxygen limitation) [83,95]. Ammonia shock loads lead to incomplete nitrification, resulting in decreased nitrogen removal efficiency, $NO_2^-$ accumulation and $N_2O$ formation. Thus, the performance of the wastewater treatment plant also influences $N_2O$ production. Oxygen limitation during nitrification could result in $NO_2^-$ accumulation and $N_2O$ formation (nitrifier denitrification pathway).

Bacterial metabolism likely necessitates a period of adjustment to adapt to shifts in process conditions, leading to significant peaks in nitrous oxide emissions. Variations in bioreactors have also been observed, e.g., a decline in DO concentration owing to elevated influent loading or aeration rate limitation [95,96], resulting in increased $N_2O$ production through the nitrifier denitrification pathway (see Section 2.3). Furthermore, the transition from anoxic conditions to aerobic conditions with the presence of accumulated $NH_4^+$ resulted in $N_2O$ formation, suggesting that the hydroxylamine oxidation pathway is an important contributor in the formation of $N_2O$ [31,70,97].

### 3.4. Substrate Composition and COD/N Ratio

The influent characteristics of a WWTP affect nitrous oxide emissions. The composition of different organic substrates is a major factor contributing to $N_2O$ emissions. Limiting availability of biodegradable organic carbon hinders complete denitrification, resulting in $N_2O$ accumulation [98,99], whereas excess carbon decreases $N_2O$ production. Influent nitrogen plays an important role in $N_2O$ emissions [5,6]. Thus, the COD/N ratio is an important factor controlling $N_2O$ production.

Several researchers have investigated the effect of organic substrates on $N_2O$ emissions on the basis of experiments conducted at lab-scale using methanol, sodium acetate and mannitol as carbon sources [100–102]. In the study of Song et al., 2015, lower $N_2O$ emissions were observed in the case of acetate compared to those emitted when using methanol as the carbon source. The $N_2O$ emission factor was 2.3% of influent nitrogen for the methanol

and 1.3% of influent nitrogen for the acetate, which was attributed to the fact that the biomass became more abundant in bacteria capable of reducing $N_2O$ with acetate as carbon source [100]. Conversely, in the study of Adouani et al., 2010, the results showed that the highest $N_2O$ and NO emissions were generated when using acetate as the carbon source, which was attributed to the diversity of denitrifying bacteria and their distinct metabolic pathways towards the added carbon substrates. The use of mannitol instead of sodium acetate as a carbon source resulted in lower $N_2O$ conversion rates (21% for mannitol and 41% for sodium acetate) [102]. Microbial analysis showed that mannitol lowered the $N_2OR$ enzyme inhibition caused by the high nitrite concentration in the partial nitrification system, thus enhancing heterotrophic denitrification. The presence of trace metals (e.g., Fe(II), Fe(III), Cu(II)) and other compounds in the substrate may affect $N_2O$ emissions through abiotic reactions [62,69,70,76]. $N_2O$ accumulation can be observed in the absence of sufficient Cu in natural waters or in the formation of nonbioavailable complexes with copper [103,104].

Increased $N_2O$ emissions were observed during denitrification when the availability of biodegradable organic carbon was limited [98,99]. $N_2OR$ is less competitive under limited COD, leading to $N_2O$ production [94,105]. Schalk-Otte et al., 2000 conducted experiments in a pure culture, observing that when the availability of organic carbon became limited, 32–64% of the nitrogen load was emitted as nitrous oxide [106]. The various denitrification enzymes (NaR, NiR, NOR and $N_2OR$) compete for electrons when conditions of limited carbon sources are favored. NaR and NiR have relatively higher affinity for electrons than NOR and $N_2OR$ [107], resulting in incomplete denitrification and $N_2O$ formation. Another cause for increased $N_2O$ emissions under conditions in which organic carbon is limited is the microbial consumption of internal storage compounds [6]. Nitrous oxide production is reduced when excess carbon is provided for the removal of electron competition [18].

Regarding the effect of different COD/N ratios (1.5, 2.5, 3.5 and 4.5) on $N_2O$ emissions, it was shown that the highest $N_2O$ production was obtained under the lowest COD/N ratio in laboratory-scale experiments [108]. Similar observations were made in the studies of Itokawa et al., 2001 and Andalib et al., 2018, where the highest $N_2O$ production was reported when the COD/N ratio was below 3.5 [109,110]. Thus, it can be concluded that low COD/N ratios correspond to high nitrogen load in the influent, resulting in elevated $N_2O$ production. Moreover, according to Law et al., 2012a, for complete denitrification, a COD/N ratio above 4 is required, with the optimal ratio ranging from 4 to 5 [111]. In agreement with the above statements, Gruber et al., 2021 demonstrated a weak positive correlation between C/N ratio and the $N_2O$ emission factor [112]. However, Quan et al., 2012, employing three lab-scale aerobic granular SBRs, reported that lowering the nitrogen loading rate or, equivalently, raising the COD/N ratio did not hinder the heterotrophic denitrification process [113].

### 3.5. pH and Temperature

One of the major factors affecting nitrification in wastewater treatment is pH. Nitrification systems are sensitive to variations in pH [114]. Wastewater biological nitrification processes are accompanied by DO consumption and pH reduction. Although the optimal pH range for complete nitrification varies between 7.5 and 8 [28,30], the optimum pH for AOB and NOB growth ranges from 8.5 to 8.8 and from 8.3 to 9.3, respectively [115]. Thus, the activity of AOB and NOB can be affected by changes in pH [116], and pH can also cause changes in the concentrations of free ammonia (FA) and free nitrous acid (FNA) (Equations (11) and (12)).

$$NH_4^+ \leftrightarrow NH_3 + H^+ \tag{11}$$

$$NO_2^- + H_2O \leftrightarrow HNO_2 + OH^- \tag{12}$$

High pH shifts the equilibrium to FA, which is the substrate of AOB [117], and is inhibitory to nitrite-oxidizing bacteria (NOB) [118,119]. The ranges of FA concentrations

that begin to inhibit nitrifying organisms are: 10 to 150 mg/L for AOB and 0.1 to 1.0 mg/L for NOB [118]. Low pH increases the FNA concentration, which inhibits both AOB and NOB [4,120]. The inhibition of both AOB and NOB was initiated at concentrations of FNA between 0. 22 and 2.8 mg/L [118].

During nitrification, the highest $N_2O$ production was observed at the lowest applied pH (pH = 6.0) [81]. NOB are strongly affected by low pH values (no activity was detected at pH =6.5) [121], thus resulting in the accumulation of nitrites. No inhibition was observed at high pH values (the activity was nearly the same for the pH range 7.5–9.95) [121]. In the partial nitrification (nitritation) process at DO = 0.7 mg/L, accumulation of nitrites was observed at high pH (pH = 7.85) [122], thus resulting in high $N_2O$ production.

During denitrification, $N_2O$ formation was observed at pH below 6.8 [123]. Similar observations were made by Hanaki et al., 1992, showing that the maximum $N_2O$ emissions occurred when pH decreased from 8 to 6.5. This was attributed to $N_2O$ reduction rate decreasing at low pH, resulting in $N_2O$ accumulation [124].

Wastewater temperature plays a significant role during nitrification. The mass transfer, chemical equilibrium and growth rate of both AOB and NOB are affected by temperature [116], and it could consequently be a major factor influencing nitrous oxide emissions. The temperatures at which the growth rates of AOB and NOB are maximized are 35 °C and 38 °C, respectively [125]. Van Hulle et al., 2007 suggested that the optimal temperatures for partial nitrification range from 35 °C to 45 °C [126]. However, only short-term effects on temperature were studied. Prolonged exposure to temperatures higher than 40 °C is likely to result in deactivation [127]. Hellinga et al., 1998 reported that at temperatures higher than 25 °C, the AOB specific growth rate increases and becomes higher than that of NOB. NOB can be washed out in activated sludge processes operating with high temperatures (30–35 °C), leading to the accumulation of nitrites and elevated $N_2O$ emissions due to the nitrifier denitrification pathway [128].

It is known that denitrification rates increase with increasing temperature [129]. It is also known that increasing temperature decreases the solubility of $N_2O$. With a temperature increase from 25 °C to 35 °C, a reduction in the solubility of nitrous oxide in water of 23% was observed [130]. Thus, nitrous oxide solubility plays a key role in controlling nitrous oxide emissions [131]. A low nitrous oxide solubility at elevated temperature leads to more $N_2O$ leaving the liquid phase before complete denitrification can be accomplished. Increasing the temperature from 10 °C to 20 °C leads to higher $N_2O$ emissions (a 2.5-fold increase was measured in nitrous oxide emissions) [36]. Poh et al., 2015 investigated the impact of temperature on nitrous oxide emissions during denitrification. The specific reduction rates of nitrates, nitrites and nitrous oxide increased by 62%, 61% and 41%, respectively, when the temperature was increased from 25 °C to 35 °C. At 35 °C, although a higher $N_2O$ reduction rate was observed, $N_2O$ became less soluble in the mixed liquor, meaning that stripping was occurring more intensively. Thus, the dissolved $N_2O$ was found to decrease continuously during the experiment because the stripping was occurring faster. As a result, although high temperatures are employed to increase the denitrification kinetics, they are expected to produce more emissions in the end.

The climate zone and the location of the WWTP also affect $N_2O$ emissions. Gruber et al., 2021 attributed the difference in $N_2O$ emissions factors between a WWTP in Finland and a WWTP in Switzerland to the influent characteristics and the climate zone. At the Viikinmäki WWTP in Finland, the seasonal dynamics in wastewater temperature are extremely strong due to the snow melting season occurring in March–April (8.8 °C). Reaction rates are known to slow down with decreasing temperature, thus resulting in low $N_2O$ emissions [132].

## 4. $N_2O$ Sampling Strategies and Measurement Techniques

Since 1980, a variety of different methods for performing $N_2O$ measurements in WWTPs have been developed [5,9,18,24,36,89,108,133–137], while various sampling strate-

gies have been used to facilitate either dissolved and/or gaseous $N_2O$ measurements [22,23,25,26,40,138–140].

According to literature, different sampling strategies have been employed for the quantification of $N_2O$ emissions. Taking samples online for twenty-four hours or online for one week and grabbing a single sample are sampling strategies used in WWTPs. The large variability in $N_2O$ emissions can be attributed to differences in the sampling methods, whereas real differences in emissions are attributable to the plant performance and show both diurnal and seasonal fluctuations [141]. Gas sampling locations depend on the tank geometry and the type of aeration system. The sampling point chosen should cover an area of at least 2% of the total tank surface [142].

Sampling protocols should be developed based on the configuration of the WWTP (e.g., the choice of sampling points). The grab-sampling technique used for gaseous fluxes does not always include sampling performed during weekends or at night, resulting in an inadequate representation of the daily fluctuations in $N_2O$ emissions [143]. Furthermore, short-term monitoring campaigns are often unable to record the daily $N_2O$ fluctuations [132,144], leading to the underestimation of $N_2O$ emissions. The emission factors assessed (0.8–2.9% of the total nitrogen loads) during long-term monitoring campaigns have been shown to be substantially higher than those found by many short-term campaigns [112]. Daelman et al., 2013b suggested that in order to quantify $N_2O$ emissions accurately, it is necessary to employ long-term or grab-sampling monitoring campaigns that take into account seasonal temperature fluctuations [145]. In the study by Gujer 2007, in which a long-term monitoring campaign was performed, the seasonal emission patterns were characterized by elevated $N_2O$ emissions during the spring season and reduced emissions during autumn [146], whereas in the study of Vieira et al., 2019, elevated $N_2O$ emissions were reported during autumn and reduced emissions during spring in a short-term monitoring campaign. The above discrepancy can be attributed to the different operational conditions and microbial species present in each process. There have been fifteen continuous long-term monitoring campaigns with 1 year duration [12,26,60,112,132,144] in which the emission factors were elevated compared to those reported in short-term campaigns [12,60,132,143,144]. Many studies have used default values referred to as emissions factors for the estimation of $N_2O$ emissions estimation [6,18,147].

Emission factors are defined as the amount of $N_2O$ emitted relative to the influent nitrogen load. The current default $N_2O$ emission factor, proposed by IPCC 2023, is 1.6% of the influent nitrogen [2]. Studies have reported that there are wide variations in $N_2O$ emission factors in lab- and full-scale WWTPs. Law et al., 2012a demonstrated emission factors ranging from 0% to 25% in different full-scale WWTPs. This variation can be attributed to the different process configurations and dynamic conditions in each case. Ahn et al., 2010 reported emission factors from 0.01% to 1.8% relative to the influent TKN. Foley et al., 2010, investigating seven full-scale WWTPs with various configurations, concluded an $N_2O$ emission factor range of 0.6–25.3% relative to the denitrified nitrogen. Bai et al., 2023 determined an emission factor of 0.09% (gr $N_2O$-N/gr TN removed) [148]. Hanaki et al., 1992, conducting lab-scale experiments, reported emission factors ranging from 0 to 8% of the nitrogen load. The different quantification methods and sampling strategies used in each process influence the $N_2O$ emission factors.

The lack of standardized measurement methods and the difficulties associated with quantifying $N_2O$ emissions from full-scale WWTPs have led to estimations of $N_2O$ emissions based on models without the input of measured data [6]. For this reason, different empirical methods have been developed to estimate and evaluate $N_2O$ emissions in WWTPs by applying fixed (default) emission factors [149–153]. According to literature, the application of fixed emission factors does not respond to the large variations in $N_2O$ emissions, may underestimate emission levels, and does not take into account the impact of the different process conditions and plant configurations [154]. In recent years, calibrated mathematical models adopting real data concerning $N_2O$ production in WWTPs have been developed for investigating the factors favoring $N_2O$ emissions in WWTPs. These models

constitute extensions of the ASM model that focus on one or more possible pathways of $N_2O$ formation, but there is a great variety in their structure depending on the number of pathways included, and their stoichiometry, kinetics, process configuration and operational characteristics [155–157]. This problem, together with the lack of real data for model calibration, complicates model application and makes the comparison of results difficult. The standardization of measurement methods, and the improvement of instrumentation for accurate measurements, together with long-term monitoring data, can make models a useful, reliable and widely applied tool for the minimization of $N_2O$ emissions [21,154].

### 4.1. Gas $N_2O$ Sampling and Measurements

In full-scale WWTPs, the $N_2O$ emissions from activated sludge tanks are usually captured using a closed floating chamber. This technique was first used to measure fluxes from non-aerated liquid surfaces [134]. The size of the floating chamber varies in different studies [9,25,142]. Spinelli et al., 2018 used two different types of gas chamber—fixed and floating—for gas collection. The fixed chambers were attached to the external wall by steel clamps and the floating chamber was fastened by ropes in order to minimize the effect of turbulence. The samples can be taken from the chambers in three different ways: (a) the outlet of each chamber can be connected through a gas tube online to an infrared gas analyzer or photoacoustic (PA) spectroscopy detection system [9,26,137,158]; (b) gas samples can be grabbed from the chamber headspace into 20 mL nylon syringes at specific time intervals and the nitrous oxide analyzed using Gas Chromatography (GC) [6]; and (c) gas samples can be obtained with suitable sampling bags, connected through a valve to the floating chamber and $N_2O$ analyzed via Gas Chromatography [142,159,160]. Moreover, $N_2O$ can be directly measured with a gas $N_2O$ sensor (Unisense A/S, Denmark) attached to the chamber [23].

In lab-scale WWTPs the off-gas can be gathered at the sample point approximately 2 cm below the top of the reactor, using a 100 mL gas syringe and $N_2O$ analyzed by GC [61]. Off-gas can also be collected continuously from the lab-scale reactor headspace and conducted via a gas tube to the online infrared analyzer [36]. Furthermore, $N_2O$ can be collected from the head of a reactor by a gas-collecting bag and analyzed via gas analyzer or GC [40].

### 4.2. Measurement of Dissolved $N_2O$

The dissolved $N_2O$ concentration can be measured in situ using an electrochemical sensor (Unisense Environment A/S, Aarhus, Denmark) placed in the center of or next to the chamber [9,22–26,74,147]. It can be also determined in liquid samples based on the salt-induced stripping approach using gas sampling bags [161]. The salt-induced stripping methodology is based on the $N_2O$ solubility in water, which is caused by the formation of a hydration shell that surrounds the gas molecules. When ionic compounds are added to a sample containing $N_2O$, the ion-dipole forces result in the salting out of $N_2O$. When the transfer of $N_2O$ from the liquid phase to the gas phase is complete, it can easily be quantified using GC. Another method for the measurement of dissolved $N_2O$ in liquid samples is its extraction in the gas phase using the equilibration headspace technique [137,142,159,162,163]. The liquid sample is sealed into a glass vial or a 50 mL syringe. After shaking, the vial and the syringe are left at room temperature for one hour without stirring. The resulting gas phase in the headspace is collected and analyzed by GC. Finally, the dissolved $N_2O$ concentration can be indirectly determined using mass balance calculations [36].

The monitoring campaign, the different sampling strategies, the measurement techniques, the process configurations, and the results obtained from each process are gathered and presented in Table 2.

**Table 2.** The monitoring campaign, the different sampling strategies, the measurements techniques, the process configurations and results obtained from each process.

| Monitoring Campaign | Sampling Strategy | Measurement Techniques | Process Configurations | Results | References |
|---|---|---|---|---|---|
| | A floating chamber was employed for off-gas collection. | Two photoacoustic (PA) spectroscopy systems were tested, one employing a quantum cascade laser (QCL) source and one using a distributed feedback (DFB) laser source for gas-phase measurements. Dissolved nitrous oxide was measured by an electrochemical nitrous oxide sensor (Unisense Environment A/S, Denmark). | A full-scale WWTP performing a two-stage deammonification process. | $N_2O$ concentration in the gas phase: 200–800 ppm. $N_2O$ concentration in the liquid phase: 1–6 mg/L. | K.M. Thaler et al., 2017 [164] |
| | Off-gas collection was conducted from the reactor headspace. Data were logged every minute for long periods of time (4–12 h of nitrous oxide monitoring). | Nitrous oxide was measured in the off-gas via an online analyzer. Dissolved nitrous oxide was not directly measured, but calculated with mass balances. | Lab-scale granular sludge airlift reactor performing partial nitritation. | $N_2O$ concentration in the off-gas: $0.08 \pm 0.01$ mg/L at 10 °C, $0.09 \pm 0.02$ mg/L at 15 °C and $0.18 \pm 0.01$ mg/L at 20 °C. $N_2O$ concentration in liquid phase: $0.12 \pm 0.02$ mg/L at 10 °C, $0.11 \pm 0.04$ mg/L at 15 °C and $0.24 \pm 0.03$ mg/L at 20 °C. | C. Reino et al., 2017 [36] |
| | Nitrous oxide samples collection in gas bags (GSB-P/0.5). | Liquid $N_2O$ was measured based on the salt-induced stripping approach and analyzed in GC. | Lab-scale reactor. | $N_2O$ emissions: $0.22 \pm 0.01$ kg/m$^3$– $1.14 \pm 0.05$ kg/m$^3$ based on the use of different inorganic salts. | P. Kosse et al., 2017 [161] |

**Table 2.** *Cont.*

| Monitoring Campaign | Sampling Strategy | Measurement Techniques | Process Configurations | Results | References |
|---|---|---|---|---|---|
| 52 days | The aerobic tank headspace was chosen as the sampling point. Fixed and floating chambers were employed for gas collection. The monitoring duration for each chamber was seven days. | Off-gas $N_2O$ was measured using an online analyzer. | Full-scale modified Ludzack–Ettinger plant. | The average $N_2O$ emission rate was $0.856 \pm 0.905$ g$N_2O$/h when the COD:TN was 3.2, while it increased to $1.850 \pm 0.972$ g$N_2O$/h when COD:TN ratio was 1.9. | M. Spinelli et al., 2018 [158] |
| Four days in June 2016 | A floating hood was employed for gas sample collection. The hood was placed in 12 different positions. One-liter Tedlar bags were used for off-gas sample collection. Gas syringes were used for dissolved $N_2O$ collection after the extraction of liquid samples. | GC with BID detector (Shimadzu 2010 Plus Tracera) was used for $N_2O$ measurements in gas and liquid phases. Dissolved $N_2O$ was extracted with the equilibration headspace technique. | Municipal water resource recovery facility with a modified Ludzack–Ettinger configuration, consisting of a series of anoxic and aerobic reactors. | Dissolved $N_2O$ concentrations varied between 9.09 and 346.9 µg/L at the different sampling points. $N_2O$ emissions varied between 2.56 and 34.99 µg/L. | D. Caniani et al., 2019 [142] |
| | Gas sampling bags (50 mL) were used for gas collection. During gas sampling, a closed gas-tight chamber with a gas-sampling outlet was fixed to the tank. | The $N_2O$ concentrations were analyzed using a gas chromatograph (Agilent 7890B, USA) with an electron capture detector (ECD). Microelectrode analysis was conducted for $N_2O$ measurements using microelectrodes (Unisense MM-Meter, Denmark). | Lab-scale SBR. | $N_2O$ emissions varied between 0 and 0.8 mg/h and 0 and 0.5 mg/h at DO = 1 mg/L and DO = 2 mg/L, respectively. Dissolved $N_2O$ concentrations varied between 0 and 0.9 mg/h and 0 and 0.3 mg/h at DO = 1 mg/L and DO = 2 mg/L, respectively. | X. Yan et al., 2019 [160] |

**Table 2.** *Cont.*

| Monitoring Campaign | Sampling Strategy | Measurement Techniques | Process Configurations | Results | References |
|---|---|---|---|---|---|
| | | $N_2O$ concentration was determined using gas chromatography (GC) equipped with an electron capture detector (GC-6890N, Agilent, USA). | Biological aerated filter. | $N_2O$ emission rate varied between 4 and 8 mg/h. | M. Zheng et al., 2019 [139] |
| | Gaseous $N_2O$ was collected from the head of sequencing batch biofilm reactors and sealed in a gas-collecting bag. | $N_2O$ off-gas was measured using the $N_2O$ detector (HA80–$N_2O$, China). The $N_2O$ concentration in the liquid phase was calculated based on equations reported by Kong et al., 2002. | Laboratory-scale sequencing batch biofilm reactors (SBBRs). | $N_2O$ emissions were $0.0741 \pm 0.0025$ mg/(L·h) with one-time dosing, whereas $N_2O$ emissions were $0.0402 \pm 0.0016$ mg/(L·h) with methanol step dosing. | H. Chai et al., 2019 [40] |
| Eighty days | | Dissolved nitrous oxide was monitored online using electrochemical $N_2O$ sensors (UNISENSE A/S, Århus, Denmark). Off-gas nitrous oxide measurements were conducted continuously and logged on a minute basis (Teledyne API, San Diego, USA). | Intermittently fed lab-scale sequencing batch reactor (SBR). | Off-gas $N_2O$ concentrations varied between 0 and 0.4 mg/L and 0 and 0.3 mg/L at pH = 8 and pH = 7, respectively; dissolved $N_2O$ concentrations varied between 0.3 and 1 mg/L and 0.1 and 1 mg/L at pH = 8 and pH = 7, respectively. | Q. Su et al., 2019 [62] |
| Two campaigns, one in summer 2015 and one in winter 2016. Each campaign lasted for one week. | A floating chamber was employed for gas collection. The gas $N_2O$ microsensor was placed in the anoxic and aeration reactor in each WWTP. The liquid-phase $N_2O$ microsensor was located close to the gas sensor. | Off-gas and dissolved $N_2O$ was measured by online $N_2O$ microsensors. | Three full-scale WWTPs, one performing a Modified Ludzack–Ettinger process, one CAS system with five plug-flow parallel reactors and one secondary treatment employing an anaerobic/anoxic/oxic (A2/O) configuration. | For the 1st campaign (summer 2015): Dissolved $N_2O$ concentrations ranged from 0.04 to 190.21 mg/L/day and $N_2O$ emissions ranged from 1.16 to 22.48 g/day. For the 2nd campaign (winter 2016): Dissolved $N_2O$ concentrations ranged from 0.04 to 353.63 mg/L/day and $N_2O$ emissions ranged from 0.21 to 492.14 g/day. | A. Vieira et al., 2019 [23] |

**Table 2.** *Cont.*

| Monitoring Campaign | Sampling Strategy | Measurement Techniques | Process Configurations | Results | References |
|---|---|---|---|---|---|
| Two campaigns, one for four weeks and one with a duration of three days | Floating hoods were used for gas collection. Sampling locations were chosen in order to cover the influent inlet area, the middle of the aeration tank, and the effluent outlet area. | The collected gas measurements were conducted using an $N_2O$ analyzer (Horiba VA-30 0 0 & VS-30 02). The $N_2O$ analyzer measurements were checked independently by analysing gas samples using a Shimadzu GC-9A equipped with a micro-electron capture detector (ECD) and a flame ionization detector (FID). The dissolved nitrous oxide was measured based on the equilibration of gas and liquid phases. | A full-scale sequencing batch reactor (SBR). | $N_2O$ fluxes range from 0 to 0.8 gr/hr*m² at different DO concentrations, whereas the dissolved $N_2O$ concentrations ranged from 0 to 0.06 mg/L. | H. Duan et al., 2020 [137] |
| Four months (January 2019–April 2019) $N_2O$ monitoring with gas analyzer lasted for March to April 2019) | | Dissolved nitrous oxide measurements were conducted employing an electrochemical sensor. Off-gas nitrous oxide emissions were continuously monitored using an MIR9000CLD gas analyzer. | A full-scale SCENA (Short-Cut Enhanced Nutrient Abatement) SBR. | $N_2O$ emissions ranged from 0 to 6 mg/m³ and dissolved $N_2O$ concentrations ranged from 0 to 2.5 mg/L. | V. Vasilaki et al., 2020 [143] |
| 27 March to 5 April of 2012 and 5th–9th of March 2013 | There is an air duct above each of the covered anoxic, aeration, and membrane tanks to directly measure $N_2O$. | $N_2O$ emissions measurements were conducted via a gas nitrous oxide analyzer. | A full-scale anoxic/aerobic membrane bioreactor. | $N_2O$ flux was 1.11 g/d in the anoxic tank, 1.03 g/d in the aerobic tank and 0.2 g/d in the membrane tank at DO = 0.5 mg/L, whereas $N_2O$ flux was 0.63 g/d in the anoxic and the aerobic tank at DO = 3.5 mg/L. | J. Wen et al., 2020 [138] |

**Table 2.** *Cont.*

| Monitoring Campaign | Sampling Strategy | Measurement Techniques | Process Configurations | Results | References |
|---|---|---|---|---|---|
| Twenty-two campaigns (2012–2018), monitoring twice, once in winter and once in summer | Aeration tanks monitoring for 1 week, including weekends. Cascade tanks monitoring in consecutive weeks. The off-gas was continuously sampled using a custom-made floating gas hood (1 $m^2$ surface area; approximately 0.3 $m^3$ headspace), placed in the middle of the monitored aerated zone. A constant off-gas flowrate was pumped out of the headspace through polyvinyl chloride tubings to a unit for dust and humidity removal (PSS-5, M&C Tech Group, Germany) and then split between the $N_2O$ and $CO_2$ infrared analyzers. | The dissolved $N_2O$ measurement was conducted using an electrochemical $N_2O$ sensor, whereas off-gas $N_2O$ measurements were performed via an infrared gas analyzer. Results comparison between the gas analyzer and GC-MS was performed. | Ten full-scale WWTPs, mainly municipal, with different configurations. | Average $N_2O$–N emission factors normalized to the TKN load in the influent to the WWTP ranged from 0.002 to 1.52%. | T. Valkova et al., 2021 [25] |
| Long-term monitoring (April 2018–July 2019) | A floating hood was employed for off-gas $N_2O$ collection. Data collection once per minute for both gas and liquid samples. | Off-gas $N_2O$ was measured via an infrared gas analyzer. Dissolved $N_2O$ was measured via an electrochemical microsensor. | A full-scale SBR performing nitrification. | $N_2O$ production rates ranged from 0.02 to 0.70 kg/day Dissolved $N_2O$ concentrations ranged from 0 to 0.38 mg/L Off-gas $N_2O$ concentrations ranged from 0 to 50 ppmv. | W. B. Bae et al., 2021 [26] |
| | A beaker was used for dissolved $N_2O$ collection. An injection syringe was used for off-gas $N_2O$ collection. | The gaseous $N_2O$ was measured via GC. The dissolved $N_2O$ was measured based on the headspace method. | A lab-scale SBR. | $N_2O$ emissions in the aerobic phase ranged from 0 to 0.1 mgN/min and from 0.025 to 0.23 mgN/min when C/N ratios were 6.5 and 9.3, respectively. | R. Yang et al., 2021 [61] |

**Table 2.** *Cont.*

| Monitoring Campaign | Sampling Strategy | Measurement Techniques | Process Configurations | Results | References |
|---|---|---|---|---|---|
| | The reactor top was sealed with a gas-tight membrane for off-gas $N_2O$ collection. The gases were drawn via sampling tube to a conditioning unit and then sent to a gas analyzer. | Infrared gas analyzer for off-gas measurements. | A pilot-scale aerobic granular sludge (AGS) SBR and a pilot-scale conventional activated sludge (CAS) SBR. | $N_2O$ fluxes ranged from 0 to 0.29 $grN_2O/hr*m^2$ and from 0 to 0.3 $grN_2O/hr*m^2$ for the AGS and CAS systems, respectively. | B.J. Thwaites et al., 2021 [140] |
| Seven campaigns with one-year duration on full wastewater treatments plants | Floating chambers were utilized for off-gas $N_2O$ collection. On WWTPs with consecutive fed lanes, one or more lanes were monitored, each with three or more floating chambers per lane placed on the aerated compartments. | Infrared gas analyzer for off-gas measurements. | Seven full-scale WWTPs with different configurations including conventional activated sludge (CAS), anoxic/aerobic (AO), anaerobic/anoxic/aerobic (A2O)), alternatingly fed intermittently aerated (A/I), and sequencing batch reactors (SBR)) and biofilm systems (hybrid fixed bed and activated sludge (IFAS), fixed bed (FB)). | $N_2O$ emission factor ranged from 0.1% to 8% of the total influent nitrogen load. | W. Gruber et al., 2021 [112] |

### 5. Strategies, Novel N$_2$O Minimization Technologies and Techniques under Development

The optimization of operational conditions in order to achieve the required effluent quality while minimizing the operating costs and energy consumption of WWTPs has been investigated by many researchers [165,166]. Control and regulations of critical operational conditions, such as DO, pH, availability of carbon sources, and solids retention time (SRT) can be applied in order to minimize N$_2$O emissions. High DO and high SRT can be achieved by applying the Complete solids Retention AS (CRAS) process. CRAS is an aerobic process that can be used to achieve the longest possible SRT (up to complete retention), maintain highly aerobic conditions in the aerobic bioreactor (DO > 4 mg/L), and perform successful microbial manipulation and efficient solid/liquid separation [167,168]. New technologies should be developed for N$_2$O emissions mitigation. Algal technologies present a hopeful solution for wastewater treatment through nitrogen and phosphorus uptake [169–171], but these technologies should be studied further to reduce N$_2$O emissions. The design and evaluation of automatic control schemes is essential to improving the performance and operational management of WWTPs and reducing N$_2$O emissions [172]. The development and employment of novel optimal control/operational strategies is necessary to improve the performance of WWTPs, such as the implementation of proportional and integral (PI) controllers or fuzzy logic control (FLC) systems [128,173].

### 6. Conclusions

In wastewater treatment plants, the production of N$_2$O can be attributed to the hydroxylamine oxidation, nitrifier denitrification and heterotrophic denitrification pathways. The contribution of each pathway to N$_2$O emissions depends on the different microbial species present (AOB, AOA, NOB, anammox) and on the process conditions. There are many different methods for measuring N$_2$O and sampling strategies for N$_2$O quantification. Gaseous N$_2$O concentration is usually analyzed by GC or via an infrared gas analyzer, whereas dissolved N$_2$O concentration is usually measured in situ using an electrochemical sensor. New strategies and controlling factors should be imposed in order to mimimize N$_2$O emissions. To reduce N$_2$O emissions during the denitrification process, a sufficient carbon source (COD/N ratio above 4), a low DO and a high pH should be maintained. For complete nitrite oxidation to nitrates, high DO (above 2 mgO$_2$/L), low nitrite concentration and high pH (pH = 8) should be ensured in order to hinder N$_2$O production. For partial nitrification (the nitritation/anammox process), low DO (<1 mgO$_2$/L) and low pH (pH = 6) are required. A long solids retention time is required to prevent nitrite accumulation during nitrification. Sudden shifts in pH or DO in the reactor should be avoided. All operational process conditions should be further studied and standardized as a guide for operators with the aim of minimizing N$_2$O emissions.

**Author Contributions:** Conceptualization, L.K. and E.A.; methodology, L.K and E.A.; software, L.K.; validation, L.K.; formal analysis, L.K. and E.A.; investigation, L.K.; resources, L.K.; data curation, L.K.; writing—original draft preparation, L.K.; writing—L.K. and E.A.; visualization, L.K.; supervision, E.A.; project administration, E.A.; funding acquisition, E.A. All authors have read and agreed to the published version of the manuscript.

**Funding:** This research was funded by Region of Western Macedonia's (MIS) 5047197 project.

**Data Availability Statement:** All data are available upon request.

**Conflicts of Interest:** The authors declare no conflict of interest.

### Glossary/Abbreviations

AOB, ammonia-oxidizing bacteria; AOA, ammonia-oxidizing archaea; NOB, nitrite-oxidizing bacteria; AMO, ammonia mono-oxygenase; HAO, hydroxylamine oxidoreductase; NaR, nitrate reductase; NiR, nitrite reductase; NOR, nitric oxide reductase; N$_2$OR, nitrous oxide reductase; FA,

free ammonia; FNA, free nitrous acid; WWTP, wastewater treatment plant; GWP, global warming potential; DO, dissolved oxygen; GHG, greenhouse gas.

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
