# Peer review of "Factors Affecting Nitrous Oxide Emissions from Activated Sludge Wastewater Treatment Plants—A Review"

_resources, doi:10.3390/resources12100114_

Round 1

Reviewer 1 Report

Comments:

The Review covered wide areas and topics and well presented, however with minor to moderate corrections.

1)Line 34 - Should be 3.2% and not 3,2%. All decimal numbers should be written without comma sign.

2)Line 35, 124 and entire manuscript should written with authors name and et.al instead of referring [ref]. For eg. Line 124 should be written as ......On the contrary Anderson et.al suggested.....[35]

3)Quality of English should be proofread.

4)Table 1-shift References column to the right.

5)Equations need to be written in journal format.

Quality of English should be improved.

Reviewer 2 Report

I am really sorry but I have to suggest rejection due to a complete lack of novelty. Although this is a review paper and it's not meant to provide with novel research information, I believe that with respect to N2O emissions from activated sludge systems a lot of review works have been already published years ago when the interest emerged and this review is therefore redundant and not needed. The following reviews already talk about nitrous oxide production in a very structured way:

Law, Y., Ye, L., Pan, Y., Yuan, Z., 2012b. Nitrous oxide emissions from wastewater treatment processes. Philos. Trans. R. Soc. B 367, 1265–1277. https://doi.org/10.1098/rstb.2011.0317

Kampschreur, M.J., Temmink, H., Kleerebezem, R., Jetten, M.S.M., van Loosdrecht, M.C.M., 2009. Nitrous oxide emission during wastewater treatment. Water Res. 43, 4093–4103. https://doi.org/10.1016/j.watres.2009.03.001

If there is any gap in these two reviews, you may write a review focusing on those gaps. The only thing that could be investigated more is the impact of pH on N2O production or the impact of temperature. As for the description of the various pathways, this is totally well-known and not needed.

I am sorry because the article is certainly well written and well referenced but in itself this work could have had a lot more value if it were submitted 10 years ago. Right now, it is not needed and it wouldn't yield citations.

A comment for the introduction:

L27-35: those important statements cannot be justified with one reference each. They need more discussion and you need to have a broad picture of all the works published containing that information. For instance, the contribution of 3.2% in line 34 is not obvious and I am quite sure that not everyone agrees with that number.

Reviewer 3 Report

This mini review about the factors affecting nitrous oxide emissions from wastewater treatment plant and explained with some references. However, the manuscript needs to modify for the publication. Current manuscript was not prepared carefully and thoughtfully. An English proofreading is necessary. 

Please avoid using words from the title as the keywords- it can narrow the search area/possibility.

Too many abbreviations without explanation, add the abbreviation separately  to help readers

Add the figures/schematic representation for more clarification.

There are some grammatical and formatting issues that need further correction.

Conclusions: This section is too brief and should be detailed. Include in this section the most important results and findings to highlight the importance of this study.

Authors must need to incorporate future prospective of the presented work in the conclusion part of the manuscript.

Extensive editing of English language required.

Reviewer 4 Report

Dear authors, as a result of consideration of the submitted article, the following has been established:

1. Are the regional location of the treatment facilities, the climatic zone of the region and other characteristics, also the performance of the treatment facilities, the features of the technological scheme of the facilities, the production index in the location of the treatment facilities, the influencing factors?

2. It is required to adjust the nitrous oxide emission factor as requested in the first question

3. Requires references to existing methodologies for calculating nitrous oxide emissions

4. It is required to bring the structure of greenhouse gases in %, including nitrous oxide, for the previous year

5. Are quantitative data available for global nitrous oxide emissions?

6. Are data available on nitrous oxide emissions from disposal and treatment facilities for sludge from municipal wastewater treatment plants?

7. Is it possible to reduce the amount of nitrous oxide emissions by shutting off the capacitive structures and venting the gas for further purification?

8. Are there catalytic ways to clean the air from nitrous oxide?

9. In table 1, you need to add data on the results of measurements

Reviewer 5 Report

Thank you for inviting me to review the article “Factors affecting nitrous oxide emissions from activated sludge wastewater treatment plants-A review”. It reviews the role of multiple factors (DO, temperature, pH, etc.) that influence NOx emissions from WWTPs. Some points should be considered before final acceptance:

1-  Don’t repeat words in title similar to the keywords.

2-  Insert a table for the design criteria (HRT, F/M, SRT, etc.) of wastewater treatment units maintaining nitrification/denitrification

3-  A figure for the nitrification/denitrification schematic process can be added

4-  Re-check the spelling and typo throughout the manuscript

5-  Add a section for the microbial species responsible for these processes

6-  The collected data should be supported by statistical analysis.

7-  The study should illustrate the removal mechanisms of C, and N. Use these articles for support https://doi.org/10.1016/B978-0-12-824270-4.00006-7; https://doi.org/10.1007/978-3-030-13913-1_19; https://doi.org/10.1007/978-981-10-3084-0_18;

8-  What about the ANAMMOX and its associated technologies?

Extensive editing of English language required

Round 2

Reviewer 2 Report

I have reconsidered my first decision. Overall, it's an appreciable work, I miss a bit of criticism in terms of what's missing from N2O research? what are the gaps? What are the contradicting factors? Is there any overlapping research?

L154 "or NO2-nitrites as the electron donor": nitrite as electron donor? What do you mean? Nitrite can be used as electron acceptor for hydroxylamine oxidation, not as electron donor. Only NOB will use nitrite as electron donor with O2 as electron acceptor.

L301-302 "accompanied by N2O formation through intense gas stripping": N2O formation does not occur through intense gas stripping. It doesn't make any sense. N2O stripping is intensified by intensified turbolence caused in turn by increased aeration. Please amend that sentence

L311-312 "Nitrites are produced during the ammonia oxidation by AOB during nitrification while nitrites are produced...": the English is wrong. You may amend as follows "Nitrites are produced both during the ammonia oxidation.... and during...."

L348-351: you have to restructure the concept of that paragraph. In general, carbon should be enough. There are maybe temporary instances of low carbon availability.

I would couple Sections 3.4 and 3.5 altogether using a more comprehensive title

L382-384: That sentence is totally meaningless. Please make a better source identifying which carbon source may or may not promote/hinder N2O emissions

Usually dissolved oxygen is indicated with DO and not with D.O. Remove the dots

L387-388: that pH ranges is too ample and lacks references. In general, the pH in a nitrification compartment may vary between 6.5 and 7.5.

L399-400: what was the reason attributed to explain the increased N2O production at increased pH?

L410-411 " Low pH inhibits the N2O reduction than on the other steps of denitrification": the "than on" is not correct English. Please rephrase

L416: "optimal" for what? You mean the temperature maximizing AOB and NOB growth rates?

There is a missing hypothesis about the effect of temperature which that since NOB rate is lower than the one of AOB at increased temperature, this promotes nitrite accumulation at high temperatures and N2O emissions. See this work which you don't need to cite: https://doi.org/10.1016/j.watres.2017.06.074

L427: production or emission? You were talking about emissions, then suddenly you switched to production. Be careful, they are not the same

L437-440: and how specifically did the climate affect N2O production? Otherwise that inclusion is useless

L460: underestimation or overestimation

L473-474: Emission factors can also be computed as N2O emitted per unit of removed TKN

L489-504: that inclusion like that without any positive criticism is meaningless.

The section future perspectives is quite meaningless. Strategies for N2O emission mitigation have been already proposed. See this review (to be cited): https://doi.org/10.1016/j.scitotenv.2022.158849 . Maybe algal technologies are something novel. I wouldn't title that section as future perspectives. Suggestions are: strategies for N2O emissions, novel N2O minimization technologies and techniques under development...

I miss the abiotic N2O production pathway.

Reviewer 5 Report

The authors' responses to my comments are satisfactory.

Minor editing of English language required
